# NEDD4 E3 Ligases: Functions and Mechanisms in Bone and Tooth

**DOI:** 10.3390/ijms23179937

**Published:** 2022-09-01

**Authors:** Ke Xu, Yanhao Chu, Qin Liu, Wenguo Fan, Hongwen He, Fang Huang

**Affiliations:** 1Hospital of Stomatology, Guanghua School of Stomatology, Sun Yat-sen University, Guangzhou 510008, China; 2Guangdong Provincial Key Laboratory of Stomatology, Guangzhou 510008, China

**Keywords:** ubiquitination, NEDD4 subfamily, Smurf1, osteogenesis, tooth

## Abstract

Protein ubiquitination is a precisely controlled enzymatic cascade reaction belonging to the post-translational modification of proteins. In this process, E3 ligases catalyze the binding of ubiquitin (Ub) to protein substrates and define specificity. The neuronally expressed developmentally down-regulated 4 (NEDD4) subfamily, belonging to the homology to E6APC terminus (HECT) class of E3 ligases, has recently emerged as an essential determinant of multiple cellular processes in different tissues, including bone and tooth. Here, we place special emphasis on the regulatory role of the NEDD4 subfamily in the molecular and cell biology of osteogenesis. We elucidate in detail the specific roles, downstream substrates, and upstream regulatory mechanisms of the NEDD4 subfamily. Further, we provide an overview of the involvement of E3 ligases and deubiquitinases in the development, repair, and regeneration of another mineralized tissue—tooth.

## 1. Introduction

Ubiquitination is a key post-translational modification of proteins that transmits specific signals to affect the stability, interaction, trafficking, localization, or activity of proteins. The process of ubiquitination begins when a single ubiquitin molecule is transferred to a substrate lysine residue. Ubiquitin is a highly conserved, tiny protein consisting of 76 amino acids. Seven internal lysine residues (Lys6, Lys11, Lys27, Lys29, Lys33, Lys48, and Lys63) or the α-amino group of N-terminal methionine (Met1) can be assembled to form various covalent chains or lead to mono-ubiquitination [1,2]. In addition, other post-translational modifications (e.g., phosphorylation, acetylation, and ribosylation) or modifications by ubiquitin-like modifiers (e.g., SUMO, NEDD8, and ISG15), enrich the classes of ubiquitin chains [3,4]. The different types of ubiquitin linkages increase the variety of protein ubiquitination, thus determining the diverse fates of proteins.

The process of ubiquitination involves three sequential enzymes. The E1 enzyme (E1) is a ubiquitin-activating enzyme that activates the recruited ubiquitin along with ATP hydrolysis. E2 enzyme (E2) is a ubiquitin-conjugating enzyme that transfers ubiquitin from charged E1 to its active-site cysteine to form a thioester bond. E3 enzyme (E3) is a ubiquitin ligase that interacts with ubiquitin-charged E2 to mediate specific transfers of ubiquitin to the target proteins directly or in a two-step reaction. According to the database, the human genome encodes more than 900 E3s and their adaptor proteins, but only 9 E1s and approximately 40 E2s [5]. During the ubiquitination process, E3s dictate the specific recognition of substrate proteins with their dual functions of both molecular matchmaker and catalyst, which is of great interest to researchers.

E3 structurally falls into three main classes: the really interesting new gene (RING) family, the HECT family, and the RING-between-RING (RBR) family [6]. Among them, the pivotal HECT family contains a conserved HECT domain of approximately 350 AA at its C-terminus. The N-terminal region of this HECT domain can bind to E2, while its C-terminal region possesses a cysteine site, where ubiquitin carried by E2 binds to form a thioester complex. The ubiquitin is then transferred to the target substrate, which is coupled to the N-terminal end of the HECT E3 ligase [7,8], as shown in Figure 1a. Based on the various architectures of the N-terminal domain, the HECT family is commonly grouped into three subfamilies: the NEDD4 subfamily, HECT and RCC-like domain (HERC) subfamily and other HECTs [9].

Several reviews have indicated the prominent roles of the NEDD4 family in the development and progression of many pathophysiological conditions, including cancers [10,11], cardiovascular disease [12], lung disorders [13], virus infection and other diseases [14]. In recent years, many studies have reported the involvement of the NEDD4 subfamily in bone and tooth-related biological processes. Dynamic changes in bone persist across the lifespan, both physiologically and pathologically. Disorders develop when the equilibration of bone resorption and bone formation is disrupted or when bone development is mistakenly conducted. During embryogenesis, two major routes for osteogenic lineage are responsible for bone formation [15]. In adulthood, bone remodeling is essential for maintaining bone homeostasis and integrity, which is tightly controlled and balanced by bone-resorbing osteoclasts and bone-forming osteoblasts [16,17]. Skeletal stem cells and multiple sources of mesenchymal stem cells (MSCs) have been identified as sources of osteoblast progenitor cells for osteogenesis [18,19]. Regulation of osteoblasts from lineage commitment to differentiation during osteogenesis is of great concern. Compelling evidence indicates that many transcriptional factors, signaling pathways, epigenetic mechanisms and environmental cues are involved in the regulation of bone formation [20,21,22,23]. Although tooth, as another mineralized tissue in the body, does not remodel like bone, there are some similarities in the regulation of tooth and bone cells and in the mechanisms of biomineralization during tissue development and repair [15]. Furthermore, both bone and tooth could be susceptible to many systemic disorders that disturb extracellular matrix formation or mineral homeostasis.

In this review, we provide a comprehensive overview of the role of each member of the NEDD4 subfamily, a certain type of HECT E3 ubiquitin ligase in osteogenesis. Furthermore, in tooth development, repair and regeneration, the regulatory role of the NEDD4 subfamily is summarized, and we also shed light on the potential involvement of other E3 ligases and deubiquitinases.

## 2. NEDD4 E3 Ligases and Bone

The NEDD4 subfamily contains nine mammalian members, as illustrated in Figure 1b: NEDD4-1/NEDD4, NEDD4-2/NEDD4L, ITCH, WW domain-containing E3 ubiquitin protein ligase 1 (WWP1), WW domain-containing E3 ubiquitin protein ligase (WWP2), NEDL1/HECW1, NEDL2/HECW2, Smad-specific E3 ubiquitin protein ligase 1 (Smurf1), and Smad-specific E3 ubiquitin protein ligase 2 (Smurf2). All members consist of three common functional domains: an N-terminal C2 domain (functioning in Ca^2+^-dependent phospholipid binding at membrane surfaces for protein localization and trafficking), followed by two to four WW domains (suggested to bind predominately proline-rich motifs of substrates, primarily PPxY motifs, phosphoserine/threonine residues, and other ligands), and a C-terminal catalytic domain (HECT domain, with ligase activity catalyzing the transfer of Ub from E2 to substrates) [24]. Furthermore, under normal conditions, the NEDD4 E3 ligase maintains its autoinhibited form by interaction of the C2 and HECT domains, unless specific signaling mechanisms release the C2 domain and give rise to E3 activation [25].

### 2.1. Smurf1

Smurf1 is a typical—and the most widely studied—E3 ligase of the NEDD4 subfamily associated with bone. Accumulated evidence suggests that Smurf1 is a key regulatory factor involved in bone physiology and bone phenotypes. Smurf1 plays a negative role in osteoblast differentiation and function, thus causing osteopenic phenotype. We summarize plenty of studies on the effects of Smurf1, including bone phenotypes, downstream substrates, regulatory mechanisms of Smurf1-substrate interactions, and potential clinical applications of recombinant human BMPs (rhBMPs).

#### 2.1.1. Smurf1 Is a Negative Regulatory Factor in Bone Phenotype

Previous research has found that overexpression of Smurf1 inhibited osteoblast differentiation through plasmid transfection in 2T3 osteoblast precursor cells. In in vivo experiments in transgenic mice, Smurf1, epitope-tagged by a 2.3-kb osteoblast-specific type I collagen promoter, was expressed in osteoblasts, and the mice showed reduced postnatal osteogenesis along with inhibition of osteoblast proliferation and differentiation [26]. Furthermore, Smurf1^−/−^ mice developed a notable phenotype, an age-dependent increase in bone mass, and osteoblasts with mutant Smurf1 markedly enhanced osteoblast differentiation [27]. Furthermore, a clinical case recently linked osteoporosis with a Smurf1 gene mutation, in which a 10-year-old girl was diagnosed with fractures and low bone mineral density, and a pathogenic microduplication involving the Smurf1 gene was detected by array comparative genomic hybridization [28]. More importantly, these clinical features are congruous with the phenotype of excess Smurf1 mutations in mice, which provides strong evidence for the use of transgenic mice as models for deep investigation.

#### 2.1.2. Smurf1 as an E3 Ligase Ubiquitylates Important Molecules Involved in Osteogenesis

Smurf1 has two WW domains and its C2 domain plays a key role in substrate selection and cellular localization [29]. Smurf1 has been predicted and uncovered to control a string of targeted proteins for ubiquitylation and degradation.

Smurf1 plays a crucial role in regulating bone morphogenetic protein (BMP)/transforming growth factor-beta (TGF-β) signaling via Smad signaling. BMP/TGF-β signaling has been suggested to be involved in lots of critical biological processes including the development and repair of the skeletal system. Studies have uncovered that Smurf1 can ubiquitinate and degrade three crucial components of Smad signaling—receptor-regulatory Smads (R-Smads), inhibitory Smads (I-Smads), and TGFβR/BMPR. The molecular interaction between the Smurf1 WW2 domain and Smad PPXY motifs was discovered, and the pivotal role of Smurf1-mediated ubiquitination of natural targets (e.g., Smad1, Smad5, and Smad6) allowed for the development of effective mimetic compounds for therapeutic strategies [30]. Initially, Smurf1 was described to selectively interact with and ubiquitinate Smad1/5, leading to their degradation and inactivation, which affected pattern formation in Xenopus embryos by inhibiting the transmission of BMP signals. In addition, Smurf1 could enhance the embryonic cell response to the Smad2 (activin/TGF-β) pathway [31]. Afterward, Smurf1 was found to directly interact with Smad7 and associate with TGFβR-I through the interaction between Smad7 and activated TGFβR-I, which induced ubiquitination and degradation of Smad7 and TGFβR-I [32]. Subsequently, the assisting role of I-Smads was identified. Smurf1 could associate with Smad1/5 and BMPRI via Smad6/7 and induce their ubiquitination and degradation [33]. Soon after, the relationships between them were partly verified in the cellular function of osteoblasts. Smurf1 overexpression blocked BMP2-mediated osteoblast conversion of C2C12 cells; meanwhile, introducing Smad5 into Smurf1-overexpressing cells rescued this effect. Likewise, data from Smurf1 depletion together demonstrated that Smurf1/Smad5 played an important regulatory role in BMP2-mediated osteoblast conversion [34].

In addition, plenty of substrate proteins of Smurf1 involved in osteogenesis have been identified. Zhao et al. identified that Smurf1 could interact directly with RUNX2 and mediate RUNX2 degradation in a ubiquitin- and proteasome-dependent manner in osteoblast precursor cells [35]. Subsequent data demonstrated that MAPK/ERK kinase kinase 2 (MEKK2) was a substrate of Smurf1-mediated ubiquitination, and in Smurf1^−/−^ osteoblasts, elevated MEKK2-JNK pathway activity was sufficient to enhance osteoblast activity [27]. Additionally, Smurf1 could control JunB turnover by interacting with its PY motif and targeting for ubiquitin-proteasome pathway, and study has suggested that Smurf1 plays a negative role in regulating BMSC proliferation and differentiation through this mechanism, partly explaining the increased-bone-mass phenotype of Smurf1^−/−^ mice [36]. Researchers also found that Smurf1- and Smurf2- mediated ubiquitination and degradation of β-catenin were enhanced during TNF and IL-17-induced impaired osteogenic differentiation of BMSCs [37]. Meanwhile, co-immunoprecipitation and GST pull-down assays showed that Smurf1 could associate with β-catenin [38]. In addition, a study showed that Smurf1 mediated ubiquitination degradation of substrate-C/EBPβ and inhibited the C/EBPβ-DKK1 axis, thereby playing an inhibitory role in the matrix mineralization of human osteoprogenitors in vitro [39].

In recent years, approaches for predicting potential substrates in the field of ubiquitination have advanced significantly. With regard to Smurf1, a study reported 89 potential substrates of Smurf1 using protein microarrays, and ten substrate proteins were confirmed by in vitro ubiquitination assay [40]. In more detail, O’Connor et al. reviewed a variety of tools and techniques for identifying substrate profiling of E3 ligases [41], providing the possibility for a deep investigation of potential substrates of E3 ligases.

#### 2.1.3. Regulation of Smurf1–Substrate Interactions

Accumulated research has provided an in-depth understanding of how Smurf1–substrate interactions are regulated as shown in Figure 2, including expression regulation, structure modification, induced degradation of Smurf1 and modification of its targeted substrates.

##### Expression Regulation of Smurf1

A few regulating factors, including signaling molecules, the inflammatory milieu, non-coding RNAs, and others, have been shown to affect the expression level of Smurf1 directly or indirectly.

Several classical signaling pathways and molecules have been involved in regulating Smurf1 expression. A previous study suggested that EGF increased Smurf1 expression via JNK/c-Jun and ERK/RUNX2 by binding to its promoter, and this EGF/Smurf1 axis inhibited Wnt/β-catenin-induced osteoblast differentiation of C2C12 cells by promoting proteasomal degradation of β-catenin [38,42]. In addition, endogenous TGF-β signaling negatively regulated osteogenic differentiation of mouse periodontal ligament cells (PDLCs), and treatment with the TGF-β receptor kinase inhibitor could downregulate the mRNA levels of Smurf1 and Smad6 and promote BMP2-dependent early commitment of PDLCs into hard tissue-forming cells [43]. In addition, MAPK-ERK signaling was activated in TGF-β-inhibited osteoblastic differentiation, as indicated by the upregulated expression of Smurf1 and the induced increase in RUNX2 and Smad1 degradation in the mesenchymal pluripotent cell line C3H10T1/2 and pre-osteoblastic cell line MC3T3. Thus, the ERK1/2 inhibitor U0126 may have potential of clinical utility in promoting osteogenesis in osteoporotic fracture repair [44]. Furthermore, TGFβ1 could induce the expression of Smurf1 at the transcript level and promote degradation of C/EBPβ, leading to the inhibition of matrix mineralization of human osteoprogenitors in vitro [39].

It is well known that the inflammatory environment has a significant influence on bone formation. Inflammatory bone disorder is a pressing issue that needs to be resolved. Therefore, inflammatory factors (such as TNF and NF-κB) play a considerable role. Researchers proposed that TNF inhibited osteoblast function, and one of the mechanisms behind this was an increase in Smurfs expression and the enhanced Smurfs-mediated RUNX2 degradation after TNF treatment in osteoblasts [45]. Likewise, through observations and experiments in transgenic animals and chronic inflammatory disorders, increased TNF levels led to bone loss, with underlying mechanisms concluded to be the induced expression of Smurf1 by TNF and the subsequent increased ubiquitination degradation of two essential proteins, Smad1 and RUNX2 in osteogenesis [46]. Another substrate, β-catenin was also involved in this process. The study found that pro-inflammatory cytokines TNF and IL-17 induced IKK/NF-κB activation and impaired osteogenic differentiation of BMSCs by promoting Smurf1 and Smurf2 mediated β-catenin ubiquitination and degradation [37]. Furthermore, the regulatory pathways were investigated. Results from C2C12 and primary cultured mouse calvarial cells revealed that TNF-α stimulated Smurf1 transcription and expression in an activating JNK/AP-1 and ERK/Runx2 signaling pathway-dependent manner [47]. Another mechanistic study demonstrated that an endoplasmic reticulum stress transducer, CREBH (a negative regulator of osteogenic differentiation and bone formation) induced Smurf1 expression and promoted Smad1 ubiquitination-dependent degradation, which was also observed after TNF-α treatment [48]. A study also found that CREB-regulated transcription coactivator (CRTC2), upregulated Smurf1 mRNA and protein, and enhanced Smurf1 promoter activity in TNF-α-reduced osteoblastic differentiation and BMP2-induced osteoblastic differentiation [49]. Additionally, a classic and multifunctional drug, melatonin could downregulate TNFα-induced Smurf1 expression and accordingly stabilize Smad1 protein, thus restoring TNFα-impaired osteogenesis of human MSCs. However, the specific regulatory mechanisms by which melatonin mediates the downregulation of Smurf1 is unclear [50]. In addition to osteogenesis, Smurf1 also plays a regulatory role in macrophage function. The research found that Smurf1 positively regulated macrophage proliferation and apoptosis, but negatively regulated migration via JNK and p38 MAPK signaling [51].

Non-coding RNAs have received considerable attention in recent years. Substantial evidence has shown that a group of miRNAs can target Smurf1 and moderate the effects on osteogenesis. Decreased miR-17 levels and increased Smurf1 expression were observed in PDLCs from periodontitis-affected periodontal ligament tissue, and both had an impact on the differentiation potential of PDLCs. Additionally, the results demonstrated that Smurf1 was directly targeted by miR-17 and a coherent feed-forward loop involving inflammatory cytokine levels, miR-17, and Smurf1 might elucidate the molecular mechanisms underlying the impact of the chronic inflammatory microenvironment of periodontitis on tissue-specific stem cells, PDLCs [52]. In addition, Smurf1, a direct target of miR-17, was involved in the p53/miR-17 cascade that negatively regulated age-dependent osteogenesis of BMSCs [53]. Additionally, a study also suggested that miR-15b played a positive role in regulation of osteoblast differentiation by directly targeting Smurf1 3′UTR and then decreasing the expression level, thus protecting Runx2 protein from Smurf1-mediated degradation [54]. Additionally, miR-503 directly suppressed Smurf1 expression, which might shed light on miR-503-promoted bone formation during distraction osteogenesis [55]. Another study suggested that Smurf1 was required for miR-672-5p-induced RUNX2 activation during osteoblast differentiation, and that miR-672-5p directly targeted Smurf1 and repressed its expression, thereby indirectly protecting RUNX2 [56]. The study also found that miR-195-5p achieved BMP-2/Smad/Akt/RUNX2 axis activation by targeting and suppressing Smurf1 to accelerate osteogenic differentiation of MC3T3-E1 in vitro and relieve osteoporosis progression in an ovariectomized (OVX) mouse model [57]. A recent study revealed that Smurf1 mRNA could be sequestered in the nucleus by increased lncRNA Neat1- paraspeckles upon mechanical loading, which prevented its expression. This mechanism inhibited Smurf1-mediated Runx2 ubiquitination and degradation, thus enhancing osteogenesis and bone formation [58].

Recently, the role of miRNAs loaded in extracellular vesicles has also been reported. In vitro and in vivo data demonstrated that extracellular vesicles from miR-148a-3p mimic-treated BMSCs from normal rats (BMSC-EV-miR-148a-3p mimic) overexpressed miR-148a-3p, which targeted Smurf1 for inhibition, leading to increased Smad7 and BCL2 expression. This signaling axis ultimately increased cell proliferation and osteogenic response, further alleviating osteonecrosis of the femoral head [59]. Another study also confirmed the targeted regulation of Smurf1 by miR-25 secreted by exosomes from BMSCs. Simultaneously, the ubiquitination degradation of Runx2 by Smurf1 was inhibited, which accelerated osteogenic differentiation, proliferation, and migration of osteoblasts [60]. Moreover, a study found that miR-19b-3p suppressed Smurf1 expression and then promoted osteogenic differentiation of BMSCs. In addition, miR-19b-3p-modified BMSCs in combination with a PLLA/POSS scaffold significantly accelerated bone repair in a rat model [61].

Other regulatory factors have also been reported in the literature. For example, parathyroid hormone (PTH) could increase Smurf1-mediated proteasomal proteolysis of RUNX2 to decrease the level of Runx2 in osteoblasts [62]. At the transcriptional level, a recent study showed that N6-adenosine methyltransferase METTL3, as a positive regulator of osteogenesis and inflammatory response in an LPS-induced inflammatory environment, negatively controlled the stability of Smad7 and Smurf1 mRNA transcripts via involvement of m6A-binding protein YTHDF2, thus restraining Smad (Smad1/5/9)-dependent signaling and osteoblast differentiation [63].

##### Structural Modification of Smurf1

Several small peptides containing the LIM domain could directly bind to Smurf1, affecting its interaction with substrate molecules during the cellular process of osteogenesis. Sangadala et al. found that LMP-1, an LIM domain protein, could directly interact with the Smurf1 WW2 domain to block Smurf1 from binding Smads (Smad1/5), likely regulating cellular responsiveness to exogenous BMP in clinical bone disease therapy [64]. Further evidence in modeling prediction and biochemical evidence also revealed that the molecular interaction between the Smurf1-WW2 domain and osteoinductive forms of LMP might have an inhibitory effect on Smurf1-mediated ubiquitination through interaction with the PPXY motifs of Smads [30,65]. Additionally, in the same region of LMP-1, which could directly interact with the Smurf1 WW2 domain, a unique motif was characterized by direct interaction with Jun activation-domain-binding protein 1 (Jab1), which could bind to Smad4/5/7. This multimolecular interaction greatly prevented these Smads from ubiquitination degradation and was involved in elevated osteogenic responses upon BMP treatment in C2C12 cells [66]. Moreover, LIM and cysteine-rich domains 1 (LMCD1), a positive regulator of osteogenic differentiation of BMSCs, mechanically cooperated with Smurf1 to protect RUNX2 and Smad1 proteins from ubiquitination degradation using luciferase reporter assay [67]. Similarly, the level of PINCH-1, also known as LIMS1 (LIM zinc finger domain containing 1), was increased by mechano-environment, such as extracellular matrix stiffening, as evidenced in this research. PINCH-1 could interact with Smurf1 and consequently prevent BMPR2 from Smurf1-dependent ubiquitination degradation, which augmented BMP signaling and controlled MSC fate decisions for osteogenic differentiation [68].

In addition, the E3 ligase activity of Smurf1 can be enhanced by targeted binding molecules. CKIP-1, also known as PLEKHO1, was suggested to be the first auxiliary factor to increase the E3 ligase activity of Smurf1. It specifically targeted the linker region between the WW domains of Smurf1 to augment complex assembly, promoting substrate binding and subsequent ubiquitylation. Additionally, CKIP-1-deficient mice exhibited accelerated osteogenesis due to decreased Smurf1 activity, which manifested as an age-dependent increase in bone mass [69]. A recent study demonstrated that increased CKIP-1 was associated with age-related suppression in Smads (Smad1/5)-dependent BMP signaling and a reduction in bone formation [70]. Research reported that Cdh1 could disrupt the autoinhibition of Smurf1 to promote E3 ligase activity of Smurf1. Subsequently, multiple downstream targets involved in osteoblast differentiation were inactivated, thus suggesting a potential strategy for treating osteoporosis [71]. Serine 148 (S148) of Smurf1 is a crucial site for its role in the development of bone formation and glucose homeostasis. S148 was involved in the phosphorylation of Smurf1 by AMPK, importantly the phenotype of mutated mice with a substitution mutation at S148 in Smurf1 was equally severe as that of Smurf1^−/−^ mice, in which the ubiquitination degradation of RUNX2 and insulin receptor by Smurf1 was disturbed [72].

##### Induced Degradation of Smurf1

Likewise, Smurf1 can be degraded by ubiquitination and other processes. Upon BMP4 stimulation, Trb3 dissociated from a “tail” domain in the cytoplasmic region of BMPRII, causing Smurf1 degradation. This process stabilized BMP receptor-regulated Smads (Smad1) and potentiated the downstream Smad pathway, facilitating BMP-mediated cellular responses, including osteoblast differentiation of C2C12 cells and maintenance of the smooth muscle phenotype of pulmonary artery smooth muscle cells [73]. FBXL15 forms an Skp1-Cullin1-F-box protein (SCF) ubiquitin ligase complex—SCF^FBXL^¹⁵—and study has found that it targets Smurf1 towards ubiquitination and proteasomal degradation. Moreover, blocking FBXL15 in rat bone tissues led to a decreased bone mass phenotype and FBXL15 could upregulate BMP signaling during adult bone formation by suppressing Smurf1 stability [74]. In addition, valosin-containing protein/p97, together with its adaptor nuclear protein localization 4 (NPL4), specifically recognized a typical motif existing in Smurf1 and delivered the ubiquitinated Smurf1 for degradation in an ATPase-activity-dependent manner. Importantly, Paget’s disease of bone-like syndrome-associated mutation of p97 harbored higher ATPase activity, thus facilitating this process, further decreasing Smurf1 protein levels and increasing BMP signaling [75].

##### Modification of Substrates—Smad1 and RUNX2

In addition to the E3 ligase Smurf1, substrates can also be modified to affect their interactions.

Smurf1-induced Runx2 degradation serves as a negative regulatory mechanism for the BMP-Smad-Runx2 signaling pathway, which influences osteoblast differentiation and bone formation. In one respect, BMP-2 signaling could stimulate Runx2 acetylation to inhibit Smurf1-mediated degradation of Runx2. Instead Runx2 deacetylation by HDAC4 and HDAC5 could promote its ubiquitination degradation [76]. In another aspect, in the presence of BMP-2, Smurf1-dependent RUNX2 ubiquitination degradation was replaced by a Smad1-RUNX2 combination to activate Smad6 gene transcription via RUNX2 binding to the OSE2 promoter site [77]. Additionally, miR-92a was involved in the interactive association by direct integration with the 3′UTR of Smad6 mRNA to suppress Smad6-mediated RUNX2 degradation induced by Smurf1 [78].

Smad1 is another classic substrate of Smurf1 in osteogenesis. Smad1 linker phosphorylation triggered by BMP and its receptor signals could enable Smurf1 to recognize and bind to it, which caused Smad1 polyubiquitination degradation and cytoplasmic retention, thus restricting its activity. Analogously, phosphorylation in the Smad1 linker region was inhibited by MAPK catalysis, providing feedback control. The counterbalance between BMP and FGF (MAPK pathway activators) played a synergetic role in bone formation [79]. Pin1 is a peptidyl-prolyl cis-trans isomerase that specifically binds to the phosphoserine-proline or phosphothreonine-proline motifs of proteins, such as Smads and RUNX2. In addition, a study reported that Pin1 played a critical role in bone formation and skeletal development [80]. Researchers found that Smad1 was conformationally modified by Pin1 specific binding, leading to the failure of Smurf1 interaction, thereby stabilizing Smad1 protein against ubiquitination degradation and sustaining its activation, which enabled Pin to be a candidate therapeutic target in many skeletal diseases [81]. Moreover, as a balancing mechanism, a study demonstrated that the ubiquitin-specific protease USP34 controlled osteogenic differentiation and bone formation by deubiquitinating and stabilizing Smad1 and RUNX2. Further experiments showed that depletion of Smurf1 rescued the osteogenic potential of Usp34-deficient MSCs [82].

#### 2.1.4. Smurf1 and Clinical Application of rhBMPs

Considerable evidence has indicated that inhibiting Smurf1 can overcome the limitations of clinical applications of rhBMPs. Increasing cellular responsiveness to rhBMPs has become a focus since their clinical application in inducing bone formation has been limited due to higher doses. Studies reported that compounds functioning as targeted inhibitors of Smurf1 had the potential to serve as BMP-sensitizers [83,84]. Furthermore, a study suggested that variations in intraosseous Smurf1 activity should be considered in bone anabolic strategies for age-related osteoporosis. The chalcone derivative, which effectively inhibited Smurf1 activity, could promote bone formation in mice with a normal BMP-2 level and elevated Smurf1 activity (BMP-2^n^/Smurf1^e^), while rhBMPs still exhibited poor performance [85]. After application of intracellular Smurf1 delivery (a tool for functional analysis of intracellular protein) with a commercially available reagent, rhBMP-2-treated MC3T3-E1.4 showed significant downregulated ALP activity, which might provide a clue for solving the off-target effects of rhBMP-2 [86]. Later, systems combining BMPs, bioactive material, and cells with Smurf1 silencing demonstrated positive bone anabolic effects in animal models. Membrane PLGA75:25 bioactivated systems, combined BMP2 and rMSC573 (rMSC with Smurf1 expression knocked down by means of siRNA) treatment demonstrated synergistic defect repair of approximately 85%, 8 weeks post-implantation into a rat calvarial, critical-size defect [87]. Another system, combining the primed MSCs of Smurf1 silencing with low doses of BMP2 sustainably releasing alginate scaffolds, could induce mature bone formation in the bone defect of the animal model and promote osteogenic differentiation in MSCs from osteoporotic patients. This approach dramatically reduced the dose of BMP2 with increased MSCs responsiveness to BMP2 by Smurf1 silencing, thus bypassing the dose-dependent side effects associated with the concomitant administration of BMP2 in clinical use [88]. Recently, a cell-based high-throughput screening approach for identifying modulators of E3 ligases was presented by integrating the ubiquitin-reference technique, and experimental data confirmed that the predicted compound, using RHOB as a Smurf1 substrate, could effectively block the catalytic activity of Smurf1 and achieve effects similar to Smurf1 loss in tumorigenesis [89]. This method could be a potential strategy for drug discovery.

### 2.2. Smurf2

Smurf2 has ~70% overall sequence identity with Smurf1, but one more WW domain (WW1). Moreover, the specific WW1 can function as an additional inhibitory element in the C2-HECT autoinhibition mechanism towards ligase activity of Smurf2, which differs from Smurf1 [90,91].

Data from multiple studies have supported a similar negative regulatory role of Smurf2 and Smurf1 in osteogenesis. Similar altered expressions of Smurf1 and Smurf2 and their substrates (such as RUNX2 and β-catenin) have been found and identified. Enhanced Smurf1 and Smurf2 expression and Runx2 proteasomal degradation were proposed to be mechanisms by which TNF inhibited bone formation in inflammatory bone disorders [45]. Likewise, upregulation of both Smurf1 and Smurf2 and degradation of Runx2 were described as anti-osteogenic effects produced by a pro-inflammatory environment created by TNF-α and IL-1β [92]. Additionally, the proinflammatory cytokines TNF and IL-17 mechanistically impaired osteogenic differentiation of MSCs by stimulating IKK/NF-κB activation and consequently promoting β-catenin ubiquitination and degradation induced by Smurf1 and Smurf2 upregulation [37]. In addition, Smurf2 was decreased in WFA (a medicinal herb that positively affected bone formation)-treated calvarial osteoblast cells and WFA-treated TNF-pretreated primary osteoblast cells [93].

Moreover, the factors underlying the regulatory mechanism of Smurf2 have been revealed. MiR-130a negatively regulated the protein level of Smurf2 by directly combining with its 3′UTR, thus promoting osteoblastic differentiation of BMSCs [94]. In vitro experiments in osteoblasts from OVX rats confirmed that lncRNA CCAT1 competitively bound to miR-34a-5p to prevent the degradation of its target gene Smurf2, thus negatively regulating osteoblast proliferation and differentiation [95]. Decreased expression of Smad7 targeted by microRNA-590-5p negatively regulated Smurf2-mediated Runx2 degradation, indirectly protecting and stabilizing the Runx2 protein thus alleviating suppression of osteoblast differentiation [96,97]. Furthermore, Akt induced Smurf2 phosphorylation to trigger autoubiquitination, thus decreasing the level of Smurf2 protein, and increasing the protein stability and transcriptional activity of Runx2 to promote osteoblast differentiation [98]. Additionally, TRAF4 positively modulated the osteogenic differentiation of MSCs by acting as an E3 ubiquitin ligase that mediated the K48-linked ubiquitination of Smurf2 at the K119 site and caused its degradation [99].

However, Smurf2 has been shown to have possible discrepant functions with Smurf1 in association with bone metabolism and repair. Surprisingly, Smurf2^−/−^ mice exhibited severe osteoporosis, including reduced skeletal size, bone mass, and increased bone resorption [100,101]. Mechanistically, researchers pointed out that Smurf2 could indirectly regulate osteoclasts through an osteoblast-dependent RANKL expression increase by ubiquitinating Smad3 and disrupting the interaction between Smad3 and the vitamin D receptor [100]. A previous study also found that parathyroid hormone (PTH) could increase the expression of Smurf2 to promote RANKL expression in osteoblasts and regulate osteoclast differentiation by triggering the ubiquitination and degradation of a class II histone deacetylase-HDAC4, thereby disrupting the HDAC4-MEF2c interaction to release MEF2c and transactivate the RANKL promoter [102]. Recently, transcriptome-wide expression analyses of human peripheral blood monocytes in postmenopausal women with low bone mineral density levels provided an innovative insight, indicating that the active interaction between TGFBR1 and Smurf2 in TGF-β signaling was predominant for bone resorption activities influenced by monocyte activities [103]. Thus, these studies imply that the effects of Smurf2 on osteoclasts are not negligible, as coordination between osteoblasts and osteoclasts is required for bone health and homeostasis. In particular, in contrast to the osteoporosis phenotype observed in Smurf2^−/−^ mice, a higher bone mass was observed in rhBMP2-induced ectopic bone in Smurf2^−/−^ mice than in WT mice [101]. Further in vitro experiments in which BMSCs from Smurf2^−/−^ mice were treated with rhBMP2 also presented increased osteogenesis by inducing the ubiquitination of Smad1/5 and negatively regulating BMP/Smad signaling [101], while Smurf2 had an impressive inhibitory effect on TGFβ signaling [97,104,105,106]. Together, Smurf2 exerts possible competitive regulation in a cell by degrading distinct Smads when responding to TGFβ and BMP signaling [107]. Further research is necessary to determine the role of Smurf2 in bone homeostasis in different cell types and TGF-β/BMP signaling regulation along with Smurf1.

### 2.3. WWPs and Other NEDD4s

In recent years, the positive effects of WWP1 inhibition on bone formation and repair have been the focus of research. An adapter protein, Schnurri-3, has been identified as an essential negative regulator of postnatal bone formation [108,109]. An early study found that Schnurri-3 could recruit WWP1, thus promoting WWP1-mediated RUNX2 polyubiquitination and degradation, and might underlie the mechanism of osteosclerotic phenotype with increased bone mass in mice lacking Sch3 [110]. Furthermore, WWP1 was found to be associated with inflammation-mediated osteoporosis. Reduced osteoblast differentiation of MSCs and increased JunB ubiquitination due to chronic TNF exposure were completely blocked in WWP1^−/−^ mice compared to wild-type mice [111]. Several studies have shown that WWP1 plays an important role in the negative regulation of osteoblastic bone formation, as increased bone formation rates and normal bone resorption parameters were observed in WWP1^−/−^ mice. Moreover, BMSCs from WWP1^−/−^ mice demonstrated enhanced migration and differentiation capacity due to decreased ubiquitinated JunB and CXCR-4 protein levels [112]. Importantly, a promising delivery system for siRNA therapeutics was developed in which WWP1 was silenced. In addition, treating fractures with this system significantly accelerated bone formation [113]. Subsequently, multiple molecular regulatory mechanisms involved in bone healing by WWP1 have been investigated. The study showed that WWP1 was targeted by miR-142-5p and consequently promoted osteoblast activity and matrix mineralization in favor of bone healing [114]. BMSC-secreted extracellular vesicles loaded with miR-15b, which targeted and repressed WWP1 expression, could promote osteogenic differentiation by attenuating WWP1-mediated KLF2 ubiquitination thus inhibiting the KLF2/NF-κB axis [115]. MiR-19b enriched in BMSC-derived exosomes targeted and suppressed the expression of WWP1 or Smurf2, thereby reducing KLF5 protein degradation by ubiquitination, promoting osteoblast differentiation, and facilitating fracture healing by modulating the Wnt/β-catenin signaling pathway [116]. In particular, nucleic acid aptamers are specifically tailored to inhibit protein targets and aptamer-mediated inhibition of protein ubiquitination, which has been developed as a novel therapeutic strategy. One particular DNA aptamer, C3A, selectively bound to WWP1 and inhibited the ubiquitination activity of WWP1, and was demonstrated to promote extracellular mineralization when treating SaOS-2 osteoblastic cells in vitro [117].

Considerable evidence has shown that WWP2 is present in cartilage and thus plays a crucial role in craniofacial development and osteoarthritis [118,119,120]. Since WWP2 could catalyze the mono-ubiquitination of goosecoid and influence craniofacial patterning [118], a study revealed that WWP2 positively regulated osteogenic differentiation of MSCs through mono-ubiquitination of RUNX2, subsequently augmenting RUNX2 transactivation and promoting osteoblastic activity, which was additionally enhanced by BMP signaling (ectopic expression of the constitutively active BMPRIA) [121].

Two studies from one team reported different bone volumes and bone mass parameters in Itch^−/−^ mice at different ages, while increased osteoclastic formation was congruously observed [122,123]. There were no differences in bone volumes or bone mass parameters between 3- and 6-month-old Itch^−/−^ mice and the corresponding age WT littermates [123]. However, young (1-month-old) Itch^−/−^ mice had high bone mass, but old (1-year-old) mice developed osteoporosis [122]. A possible explanation was that the aging process might positively influence the catabolic effect of Itch depletion on osteoclast-mediated bone resorption and cover the anabolic effect of Itch depletion on osteoblasts [122]. On the one hand, it has been suggested that Itch negatively regulates osteoclast formation. In vitro bone marrow cells or spleen cells from Itch^−/−^ mice exhibited increased osteoclast formation, and the regulatory mechanisms of Itch were investigated using osteoclast precursors in which Itch was targeted by RANKL and induced TRAF6 signaling. Itch deficiency led to the loss of CYLD-TRAF6 binding and reduced TRAF6 deubiquitination, promoting RANKL-induced osteoclast formation [123]. Additionally, in vitro and in vivo experiments showed Itch deficiency terminated the effects of the antidepressant Clomipramine (CLP) on osteoclast formation and bone loss [124]. On the other hand, Itch has positive effects on bone formation. Bone marrow mesenchymal progenitor cells (BM-MPCs) from Itch^−/−^ mice were suggested to have increased osteoblast differentiation and bone formation potential in vitro and in vivo, and the underlying mechanism was unfolded that JunB was ubiquitylated and degraded by the E3 ligase Itch [122]. Moreover, researchers have implied that Itch has a positive effect on bone fracture healing. A study reported that the expression levels of the NEDD4 subfamily (WWP1, Smurf1, Smurf2 and ITCH) were increased in the fracture callus of tibial fracture models compared with samples from non-fractured bones, specifically with an elevated total amount of ubiquitinated proteins and NF-κB members. Itch depletion in a mouse model of tibial fracture resulted in higher expression levels of osteoblast-associated genes, including Runx2 and ALP, in fracture callus tissue. In C2C12 cells, overexpressed RelA and RelB could bind to the NF-κB binding sites of the Itch promoter to increase Itch expression [125].

A genome-wide association study revealed that NEDD4L was the only gene that overlapped with the identified copy number polymorphism associated with height variation in Chinese females. Thus, it was suggested to be an important candidate gene in the regulation of bone metabolism [126].

In addition, one study elucidated the positive role of RSP5, also called NEDD4L, in regulating the osteogenesis of human MSCs. The underlying mechanism was revealed to be RSP5-mediated induction of K63-linked polyubiquitination of Akt, followed by Akt pathway activation [127].

## 3. NEDD4 E3 Ligases and Tooth

Tooth, especially dentin, shares many similarities with bone. A mature tooth consists of an outer enamel crown, dentin body, and cementum that covers the root. Dentin is similar to bone in several aspects, particularly in the composition of the extracellular matrix, which is secreted by well-differentiated odontoblasts and osteoblasts. Unmineralized predentin is converted to dentin when the collagen-rich extracellular matrix (ECM) becomes mineralized, which is similar to the mineralization and conversion of osteoid to bone [128]. The small integrin binding ligand, N-linked glycoprotein (SIBLING) family (e.g., OPN, BSP, DSPP, DMP1 and MEPE) is thought to play key biological roles in the initiation and control of mineralization of both bone and dentin [129]. Additionally, pre-odontoblasts derived from mesenchymal pulp progenitors can transdifferentiate into osteoblasts during tertiary dentinogenesis [130]. The two most striking examples of system disorders that affect both bone and dentin are osteogenesis imperfecta and hypophosphatemic rickets [131]. Osteogenesis imperfecta is also mainly characterized by dentinogenesis imperfecta due to the abnormal local organization of the ECM. Hypophosphatemic rickets are identified in several forms depending on their inheritance and gene mutation, and skeletal and dental hypomineralization is associated with abnormal secretory ECM proteins and calcium and phosphate metabolism.

The mechanisms underlying tooth development, repair, and regeneration have not yet been elucidated. With regard to the similarities between bone and tooth, some studies have focused on the potential involvement of ubiquitination and deubiquitination in tooth-related processes, as illustrated in Figure 3. Proteasome-dependent protein degradation is the most common outcome of ubiquitinated proteins. Data from animal experiments revealed that the addition of the classical proteasome inhibitor bortezomib to a calcium hydroxide-based material during direct pulp capping could induce more apparent dentin bridge structures and facilitate maintenance of the integrity of the remaining pulpal tissue [132]. This finding indicates that ubiquitination process may participate in the regeneration of the pulp-dentin complex.

Several members of the NEDD4 subfamily play a similar role in the odontogenic differentiation of dental-derived MSCs. TGF-β1 could induce Smurf1- and Smurf2-mediated NFIC degradation by promoting formation of the Smad3 complex during odontoblast differentiation of MDPC-23 [133]. A regulatory circuit between RUNX2 and Smurf1 was disclosed during odontoblastic differentiation of human dental pulp cells (hDPCs); RUNX2 binding to the Smurf1 promotor mediated Smurf1 expression and Smurf1 ubiquitinating RUNX2 decreased its protein level, which provided a point of view in the molecular mechanism of tooth development [134]. LncRNA MEG3 could competitively bind miR-543 to enhance Smurf1 expression, hence playing an inhibitory role in osteogenic differentiation of hDPCs [135]. Similarly, Smurf2 was targeted by miR-497-5p and influenced Smad signaling, which accounted for the positive regulatory function of miR-497-5p of the osteo/odontogenic differentiation of stem cells from apical papilla (SCAPs) [136]. Furthermore, WWP2 has recently been indicated to play a role in accelerating odontoblast differentiation of mouse dental papilla cells (mDPCs) and dentinogenesis by revealing a mechanism associated with PTEN and KLF5. A study verified that WWP2, which is more involved in substrate mono-ubiquitination, could promote odontoblastic differentiation of mDPCs through WWP2-mediated KLF5 mono-ubiquitination in the nucleus, and that the PPPSY (PY2) motif of KLF5 and cysteine 838 (Cys838) of WWP2 were crucial for WWP2-mediated KLF5 transactivation [137]. Moreover, the in vivo promotive role of WWP2 in dentinogenesis was identified in WWP2 knockout mice [138]. In addition, the results demonstrated that PTEN was targeted by WWP2 for ubiquitination degradation, and its inhibitory effect on the transcriptional activity of KLF5 was suppressed by WWP2. [138].

In addition, other E3 ligases and deubiquitinases have been identified during tooth development. A study found that E3 ubiquitin ligase Mdm2 mediated the ubiquitination and degradation of p53 and mono-ubiquitination of Dlx3 to promote odontoblast-like differentiation of mDPCs by excluding the inhibitory effect of p53 on odontoblast-like differentiation and upregulation of Dlx3 and its target gene Dspp [139]. Furthermore, in vivo Mdm2 knock-out mice exhibited defects in dentin formation, and the active interaction with substrate Dlx3 in the odontoblast nucleus was confirmed in mouse molars. However, another substrate p53 was independently associated with Mdm2-involved dentinogenesis, as indicated by the injection of a small-molecule inhibitor Nutlin-3a into wild mice [140]. Deubiquitination can reverse the process of ubiquitination, by which deubiquitinases remove ubiquitin from ubiquitin-modified proteins [141]. Conditional deletion of USP34 in the dental mesenchyme cells disturbed deubiquitination-mediated NFIC stability, thus frustrating the odontogenic potential of DPCs, and a short root anomaly was observed in USP34 deletion mice [142]. USP49 was newly identified as a deubiquitinating enzyme of two transcription factors responsible for hypodontia and oligodontia, paired box gene 9 (PAX9) and Msh Homeobox 1 (MSX1), to stabilize their protein levels. Results from in vitro and in vivo experiments verified that USP49-mediated PAX9/MSX1 deubiquitination was essential for successful odontogenesis during tooth development [143]. In addition to the development of mesenchymal cells in the tooth germ, epithelial cell-mediated enamel formation also involves ubiquitin protein ligases. The core subunit of the E3 ligase anaphase-promoting complex (APC), ANAPC10, and a member of the F-box protein, Skp2, one subunit of ubiquitin protein ligase complex SCFs (SKP1-cullin-F-box), participated in the mechanism of transcription factor Sox21-controlled epithelial-to-mesenchymal transition, which determined the fate of ectodermal organs [144].

## 4. Perspectives

Protein ubiquitination is an essential process in the cellular functions of the skeletal system. As a link between ubiquitin and substrate proteins, E3 ligases specifically recognize substrates and achieve diverse ubiquitin modifications, thus functioning as critical regulators of various cellular processes. Members of the NEDD4 subfamily have been demonstrated to play distinct roles in bone formation. To date, the role of Smurf1 has been extensively explored in bone, especially its negative regulatory effect on the process of osteogenesis, thus providing possible strategies for bone formation by Smurf1 inhibition. However, the mechanisms by which other members regulate skeleton cell functions and the actual mechanisms behind the phenotypes of knockout mice have not been exhaustively described. With more comprehensive and systematic research, drugs such as E3 ligase inhibitors and cell therapies with gene silencing will be designed to treat diseases related to hard tissues.

Although the formation of bone and tooth are similar physiological processes, and the correlation between the NEDD4 subfamily members and osteogenesis has been extensively reported, there are still few studies focusing on the tooth and ubiquitination mechanisms. We have not only summarized the roles of the NEDD4 subfamily but also highlighted the role of other E3 ligases and deubiquitinases in tooth development and dental-derived stem cell odontoblastic differentiation, thus providing new ideas for elucidating the mechanisms of tooth development, repair, and regeneration.

## Figures and Tables

**Figure 1 ijms-23-09937-f001:**
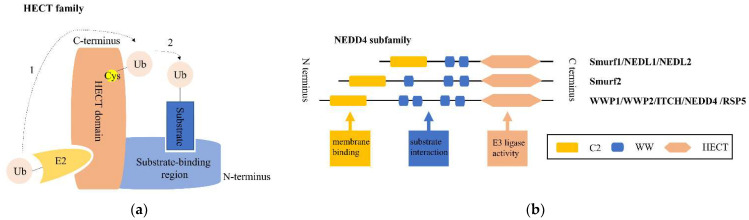
Overview of the structure of the HECT family and the NEDD4 subfamily. (**a**) The HECT family is shown combined with a ubiquitin (Ub)-conjugated E2. Dotted line 1 indicates that Ub is first transferred to the active-site cysteine (Cys) of the HECT domain, before attaching to a lysine on the substrate coupled with the substrate-binding region of the NEDD4 E3 ligase represented by dotted line 2. (**b**) The NEDD4 family generally shares three functional domain parts: an N-terminal C2 domain (yellow) for lipid membrane binding, central 2–4 WW domains (blue) involved in the interaction with substrates, and a C-terminal HECT catalytic domain (orange) associated with E3 ubiquitin ligase activity.

**Figure 2 ijms-23-09937-f002:**
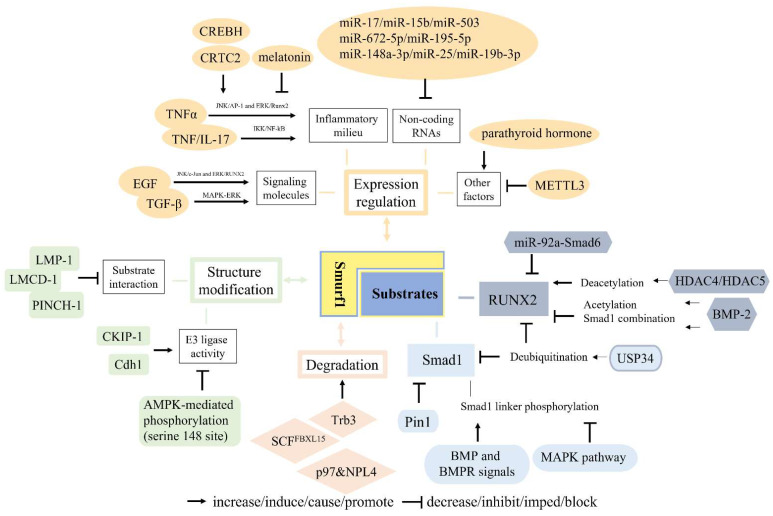
Smurf1–substrate interactions are regulated in many aspects. Smurf1 and its two common substrates, RUNX2 and Smad1, are respectively regulated. As for Smurf1, the upstream regulatory mechanisms are summarized as expression regulation, structure modification and induced degradation indicated by a double-headed arrow. Signaling molecules, inflammatory mediators, non-coding RNAs and other factors could increase or decrease Smurf1 expression. Several molecules can bind to Smurf1, thus inhibiting the interaction between Smurf1 and the substrates (LMP-1, LMCD-1, PINCH-1), or influencing its E3 ligase activity (CKIP-1, Cdh1, AMPK-mediated phosphorylation in serine 148 site). Trb3 dissociation, SCFFBXL^15^ and p97-NPL4 complex can induce degradation of Smurf1. As for substrate modification, RUNX2 deacetylation by HDAC4/5 causes its degradation, but acetylation and Smad1 combination upon BMP2 stimulation, as well USP34-mediated deubiquitination and miR-92a-targeted Smad6 suppression can impede RUNX2 degradation. Smad1 linker phosphorylation can promote Smad1 ubiquitination degradation by Smurf1; however, Pin1 binding and USP34-mediated deubiquitination block its degradation and maintain its stability.

**Figure 3 ijms-23-09937-f003:**
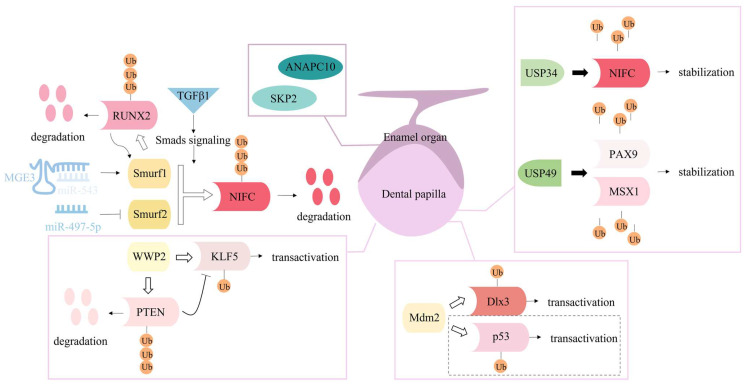
Ubiquitination and deubiquitination are crucial mechanisms in the regulation of tooth repair and development. Three NEDD4 subfamily members, Smurf1, Smurf2 and WWP2, have been found to regulate odontoblastic differentiation; their downstream substrates (RUNX2, NIFC, KLF5 and PTEN), ubiquitination types (polyubiquitination or monoubiquitination), and outcomes (degradation or transactivation), as well as several upstream regulators (non-coding RNA and signaling molecules) are displayed. Another E3 ligase, Mdm2, and deubiquitinases, USP34 and USP49 play a regulatory role in odontoblastic differentiation of dental papilla cells. The dotted box denotes the Mdm2-p53 mechanism is supported in vitro but denied by data from in vivo Mdm2 knock-out mice. Additionally, two subunits of ubiquitin protein ligase complexes, ANAPC10 and SKP2 may participate in the fate determination of enamel organs.

## Data Availability

Not applicable.

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
