# Peer review of "NEDD4 E3 Ligases: Functions and Mechanisms in Bone and Tooth"

_ijms, 2022, doi:10.3390/ijms23179937_

Round 1

Reviewer 1 Report

The manuscript entitled "NEDD4 E3 Ligases: Functions and Mechanisms in Bone and Tooth." has been submitted as a review by Xu et al.

The manuscript text is well written and the figures support the understanding of the topic. In general, this is a very informative and interesting review of a special aspect of NEDD4-family ligase functions.

Author Response

We appreciate your positive comments. We hope our work will give a significant overview and prospect to the literature associated with the functions and mechanisms of NEDD4 E3 ligases in bone and tooth. Thank you very much for your support.

Reviewer 2 Report

In their manuscript, Xu and colleagues present a literature review of the implication of NEDD4 ubiquitin ligases in bone and tooth. After a general introduction to the ubiquitin system and description of NEDD4 E3 ubiquitin ligases, they divided the core of the manuscript into two main sections: (1) roles of NEDD4s in bones and (2) roles in NEDD4s in teeth. Both sections review the role and modulation of different NEDD4s (Smurf1/2, WWPs, ITCH…). The review would interest the scientific community working on the modulation of bone and tooth development.

Some minor comments remain.

1.     The use of present and past tenses is not consistent and need to be corrected. In addition, additional proofreading of the manuscript would be required.

2.     Lines 70-72: the sentence starting with “Despite the fact that another mineralized tissue […]” is unclear and needs to be rephrased.

3.     Lines 77-79: the sentence starting with “Furthermore […]” should be rephrased. Should it be in the present tense, as in the previous sentence?

4.     Line 87: “Ca2+” -> the “2+” should be superscripted.

5.     Line 150: what do the authors mean by “on their heels”?

6.     Line 153: what do the authors mean by “super-infection with Smad5”?

7.     Lines 184, 281, 322: are these new sections?

8.     Line 204: the sentence starting with “it is well-known […]” is unclear and would need to be rephrased.

9.     Line 231: “ill defined/unveiled”?

10.  Line 605: Deubiquitination itself does not constitute a posttranslational modification.
